# The protective effect of iron isomaltoside on myocardial ischemia-reperfusion injury via the suppression of KLF4/NF-κB signaling

Huiping Gong[1], Qingyang Zhao[2], Jingbo Zhang[3], Duanchen Sun[4], Xianghua Zhuang[5], Qiaofeng Dong[6], Aixia Dou[7]*

1 Department of Emergency, The Second Hospital of Shandong University, Cheeloo College of Medicine, Shandong University, Jinan, Shandong Province, China, 2 Department of Cadres Medical Care/Gerontology Geriatric, The Second Hospital of Shandong University, Cheeloo College of Medicine, Shandong University, Jinan, Shandong Province, China, 3 Department of Cardiology, The Second Hospital of Shandong University, Cheeloo College of Medicine, Shandong University, Jinan, Shandong Province, China, 4 School of Mathematics, Shandong University, Jinan, Shandong Province, China, 5 Department of Endocrinology, The Second Hospital of Shandong University, Cheeloo College of Medicine, Shandong University, Jinan, Shandong Province, China, 6 Department of Haematology, Heze Municipal Hospital, Heze,Shandong Province, China, 7 Department of Haematology, The Second Hospital of Shandong University, Cheeloo College of Medicine, Shandong University, Jinan, Shandong Province, China

* douax0110@163.com

## Abstract

This study aimed to investigate the beneficial effects of iron isomaltoside (IIM) on myocardial function and the associated mechanisms in rats with myocardial ischemia/reperfusion (I/R)-induced damage and hypoxia/reoxygenation (H/R)-induced H9C2 cells. Changes in cardiac pathology after myocardial infarction (MI) were analyzed with hematoxylin-eosin staining. Myocardial cell apoptosis in the heart tissues of rats with MI was assessed using TUNEL staining. In H/R-induced H9C2 cells, cell viability and lactate dehydrogenase (LDH) and adenosine 5'-triphosphate levels were detected. Apoptosis and MMP in H9C2 cells were detected with flow cytometry. Our results demonstrated that IIM treatment reduced myocardial injury induced by ischemia-reperfusion (I/R) and suppressed cardiomyocyte apoptosis, inflammation, and autophagy induced by I/R in rats. Moreover, we confirmed that IIM repressed apoptosis and regulated MMP in H9C2 cells exposed to H/R. IIM relieved the inflammatory response and autophagy in H/R-treated H9C2 cells. In addition, IIM inhibited the Krüpple-like factor 4 (KLF4)/NF-κB pathway in H/R-induced H9C2 cells. Interestingly, the function of IIM on apoptosis, MMP, inflammation and autophagy were abolished by KLF4 overexpression in H/R-induced H9C2 cells. In conclusion, IIM could repress cardiomyocyte apoptosis, inflammation and autophagy through the inhibition of the KLF4/NF-κB pathway and thus reduced myocardial injury *in vivo* and *in vitro*.

**Data availability statement:** All relevant data are within the paper and its Supporting Information files.

**Funding:** This study was funded by the Bethune Foundation under the grant "Feifan Iron Supplement-Improving the Diagnosis and Treatment Capacity of Iron Deficiency Anemia" (grant number Ffbt-C 2022-010), the Shandong Medical Association Clinical Research Fund-Qilu Special Project (grant number YXH2022ZX02186), and the Key Science and Technology Program of Shandong Province (grant number 2017G006029). Role of Pr. Aixia Dou: Funding acquisition (grant number Ffbt-C 2022-010), Supervision, Writing-review&editing. Role of Dr. Qiaofeng Dong: Funding acquisition (grant number YXH2022ZX02186), Resources, Methodology. Role of Dr. Huiping Gong: Funding acquisition (grant number 2017G006029), Data curation, Investigation.

**Competing interests:** The authors have declared that no competing interests exist.

## Introduction

Myocardial infarction (MI), a rapid-onset cardiovascular disorder, carries a substantial mortality rate and frequently results in the development of chronic heart failure [1–4]. CHF is a progressive condition characterized by the inability of the heart to pump sufficient blood to meet the body's demands [5]. The treatment of MI mainly focuses on reperfusion as soon as possible via thrombolytic therapy. But paradoxically, early reperfusion therapies can induce irreversible myocardial ischemia/reperfusion (I/R) damage [6,7]. Therefore, it is crucial to investigate effective therapeutic strategies that can improve the condition of myocardial I/R injury.

Iron is a vital element involved in numerous physiological processes within the human body. Iron deficiency is a common nutritional disorder that affects a significant number of people with heart failure, affecting as many as 75% of patients [8]. Conversely, both primary and secondary iron overload can contribute to the development of heart disease by causing oxidative damage [9,10]. However, the exact underlying mechanisms involved in this process have yet to be fully elucidated. Previous studies have suggested that excessive iron in cardiomyocytes directly trigger ferroptosis by accumulating phospholipid hydroperoxides in the cell membrane [11]. Ferroptosis may subsequently contribute to myocardial ischemia/reperfusion injury [12]. Recently, there has been increasing interest in Iron Isomaltoside 1000 (IIM), a complex composed of iron hydroxide and isomaltoside, for its therapeutic potential. IIM has shown remarkable affinity for iron binding, making it a promising candidate for various applications beyond treating iron deficiency anemia [13,14]. Intravenous administration of IIM has effectively corrected anemia in patients, highlighting its clinical relevance in routine medical practice [15]. Moreover, studies have shown significant efficacy and favorable tolerability of IIM in patients with chronic kidney disease [16]. For heart failure patients with left ventricular ejection fraction (LVEF) <45%, alongside with iron deficiency symptoms, it is recommended to supplement iron intravenously to alleviate heart failure symptoms and improve quality of life. These findings highlight the potential of IIM as an exciting therapeutic option for a wide range of diseases. However, whether IIM directly impacts the underlying mechanisms of these diseases remains unknown. So we suspect that iron-binding IIM may have a protective effect on myocardial ischemia/reperfusion injury, as IIM has demonstrated significant affinity for iron.

The aberrant functionality of Krüppel-like factors (KLF) family members is intricately linked to various metabolic disorders, cardiovascular ailments, and malignant neoplasms [17]. Among these factors, KLF4, known for its pro-inflammatory properties, exerts a pivotal role in modulating the pro-inflammatory pathway of macrophages [18]. Additionally, KLF4 assumes a crucial role in myocardial hypertrophy [19]. Furthermore, extensive research has demonstrated the essential role of KLF4 in the modulation of the NF-κB signaling pathway and the subsequent orchestration of inflammatory responses. Mechanistically, the regulatory function of KLF4 has been attributed to its ability to activate Rho-related GTPase RHOF [20]. The NF-κB signaling pathway is widely acknowledged as a key pro-inflammatory pathway in various diseases, including heart failure and myocardial infarction, where it assumes a critical

role in the underlying pathophysiological mechanisms [21,22]. However, the precise regulatory mechanisms of this signaling pathway in myocardial ischemia-reperfusion injury are not fully understood.

Therefore, the main objective of this study was to evaluate the potential cardioprotective properties of IIM in both H9C2 cells and a rat model of myocardial infarction subjected to I/R or H/R conditions. Our research aimed to suppress the KLF4/NF-κB pathway and investigate its effects. H/R refers to the process of subjecting cells or tissues to a period of oxygen deprivation (hypoxia) followed by re-introduction of oxygen (reoxygenation). The results of our study unequivocally demonstrate that IIM effectively reduces apoptosis, inflammation, and autophagy in cardiacmyocytes by suppressing the KLF4/NF-κB pathway. Consequently, it significantly mitigates damage to the heart muscle both in laboratory settings and living organisms. These groundbreaking discoveries provide compelling evidence for the potential use of IIM as a therapeutic strategy in treating myocardial ischemia/reperfusion injury.

## Materials and methods

### 1. Experimental animals

Male Sprague-Dawley (SD) rats, 8 weeks old and weighing 260-300g, were obtained from Pengyue Experimental Animal Breeding Co., Ltd. in Jinan, China. The rats used in our study were specifically bred and verified to be free of pathogens. We used pentobarbital sodium 100mg/kg intraperitoneal injection to euthanize rats. The research protocol was subjected to a comprehensive evaluation and received approval from both the Ethics Committee of the Second Hospital of Shandong University. IIM was obtained from Wasserburger Arzneimittelwerk GmbH.

### 2. Myocardial ischemia-reperfusion model and grouping

An experimental model of myocardial I/R was established by ligating the left coronary artery 5mm away from its proximal end for a duration of 30 minutes, followed by releasing the ligation and allowing for 4 hours of reperfusion. The rats received single intravenous injections of IIM through the tail vein before undergoing I/R treatment. Then the animals used in this study were randomly assigned to one of the following five groups:1) Sham group (n=5), where rats underwent surgery without the ligation procedure; 2) Ischemia/Reperfusion (I/R) group (n=5), where rats underwent I/R; 3) I/R+IIM 5 group (n=5), where rats underwent I/R and were administered 5mg/kg of Iron Isomaltoside 1000 (IIM) treatment; 4) I/R+IIM 10 group (n=5), where rats underwent I/R and were given 10mg/kg of IIM treatment; 5) I/R+IIM 20 group (n=5), where rats underwent I/R and were provided with 20mg/kg of IIM treatment. The samples were collected on the 7th day following I/R injury.

### 3. Histological analysis

Left ventricular heart tissue samples from each rat were fixed in buffered paraformaldehyde, embedded in paraffin, and sliced into slices that were 5μm thick.To conduct morphological analysis, these sections were treated with a solution of hematoxylin and eosin (H&E) from Solarbio, located in Beijing, China.To perform immunohistochemistry, the heart tissue sections underwent a 15-minute treatment with 3% hydrogen peroxide to remove the activity of endogenous peroxidase. Following the washing process, the sections were left to incubate overnight at a temperature of 4°C with the LC3 antibody (1:100; #4108, Cell Signaling, USA). Afterward, the sections were exposed to the secondary antibody for a total duration of 60 minutes. Following color development with 3,3'-diaminobenzidine and counterstaining with hematoxylin, images were observed and captured using an optical microscope. Terminal deoxynucleotide transferase-mediated dUTP nick end-labelling (TUNEL) staining was used to assess apoptosis using a TUNEL Apoptosis Assay Kit (Beyotime, Jiangsu, China). Briefly, the heart section was immersed in TUNEL solution in the darkness at 37°C for 1h. Subsequently, the cell nuclei were stained utilizing 4',6-diamidino-2-phenylindole (Beyotime, Jiangsu, China). Finally, apoptotic cells were counted in a blinded fashion utilizing ImageJ software (National Institutes of Health, Bethesda, MD, USA).

## 4. Cell culture and myocardial H/R model

Rat Cardiomyocytes Cells, H9C2 cells (No. tings-52661) were purchased from Hefei Wanwu Biotechnology Co., Ltd. H9C2 cells were maintained in a controlled environment at 37°C with 5% CO2 until they reached a confluency of 80–90%. Subsequently, the cells were exposed to a hypoxic environment by incubating them in a low-glucose buffer at 37°C for 6 hours. After the hypoxic exposure, the cells were then reperfused by transferring them to an oxygenated complete culture medium.

## 5. Cell transfection

To overexpress KLF4 in H9C2 cells, we employed pcDNA3.1-KLF4 vector and mixed it with Lipofectamine 3000 to pre-pare transfection complexes. These complexes were then added to the cell culture medium for co-culturing with H9C2 cells for a defined duration.

## 6. Analysis of cell viability

The cells from each treatment group were cultured in a 96-well plate and maintained under specific experimental conditions. Afterward, CCK-8 reagent (Beyotime, Jiangsu, China) was introduced, and the culture dishes were incubated at ambient temperature for several hours to allow for colorimetric measurement of absorbance.

## 7. Detection of LDH Level and ATP and FERRITEN

LDH and ATP levels were detected separately using LDH Cytotoxicity and ATP Assay Kits (Beyotime, Jiangsu, China), respectively. Ferritin were detected by ELISA.

## 8. Detection of apoptosis and mitochondrial membrane potential (MMP)

After cell collection, we subjected them to incubation in the absence of light with Annexin V-FITC and propidium iodide (PI) for a total duration of 30 minutes. Subsequently, we employed the FACSCalibur flow cytometer from BD Biosciences to assess cell apoptosis. To detect the fluorescence signal of H9C2 cells, we cultivated the cells in a fresh culture medium and supplemented it with the rhodamine 123 working solution obtained from Sigma. Following gentle agitation of the cells in a constant temperature bath set at 37°C for 30 minutes, we loaded them into the flow cytometer and measured the emitted fluorescence signal at a wavelength of 525nm.

## 9. Immunofluorescence

To extract antigens from paraffin-embedded tissue sections (Sola Biosciences, Beijing, China), we utilized a solution containing ethylenediaminetetraacetic acid (EDTA). Following that, the sections were obstructed using 10% goat serum that lacked immunity, at a temperature of room, for a duration of 30 minutes. After that, the sections were kept at a temperature of 4°C overnight with an LC3 antibody (1 100; #4108, CST, USA), then incubated with a secondary antibody (Protein, Wuhan, China) in darkness for a duration of 60 minutes. We utilized fluorescence microscopy to observe the stained specimens after performing a 10-minute DAPI counterstaining of the cell nuclei (Beyotime, Jiangsu, China). Furthermore, H9C2 cells were treated with 4% formaldehyde for a total duration of 30 minutes and then subjected to permeabilization using 0.5% Triton X-100for a period of 20 minutes. Afterward, the cells were stained using the previously mentioned LC3 antibody.

## 10. RNA extraction and qRT-PCR

To isolate total RNA from each experimental group, we employed TRIzol reagent (Invitrogen). Subsequently, following the guidelines of the reverse transcription kit (Vazyme), RNA was converted into complementary DNA (cDNA) through reverse

transcription. Fluorescence-based quantitative PCR was conducted using a qRT-PCR reagent kit. The primer sequences specific to each gene are provided in Table 1 below.

## 11. Western blot analysis

In this study, we utilized Western blotting technique to evaluate the levels of protein expression. In brief, we initially isolated cellular proteins from various treatment groups and conducted electrophoresis and electroporation techniques. Subsequently, we incubated the samples with primary and secondary antibodies, and the protein expression levels of the target protein were detected using the chemiluminescent method. The primary antibodies employed in this study are presented in Table 2 provided below.

## 12. Statistical analyses

Data between groups was analyzed using GraphPad Prism 8.0 software (SanDiego, CA, USA). Data were analyzed using one-way ANOVA. Data are displayed as the mean ± standard deviation (SD). A statistical p-value of < 0.05 was considered significant.

## Results

### 1. IIM prevented I/R-caused MI and repressed cardiomyocyte apoptosis in rats

The histopathological analysis using hematoxylin and eosin (HE) staining revealed a pronounced myocardial cell damage in the I/R group, accompanied by evident infiltration of neutrophils (Fig 1A). Remarkably, administration of IIM exhibited a significant amelioration of this phenomenon (Fig 1A). TTC staining also confirmed the amelieration (Fig S1D). Furthermore, IIM supplementation decreased ferritin concentration in the IR+IIM group (Fig S1A), indicating no ferroptosis, we further investigated the role of the IIM in cardiomyocyte apoptosis by employing TUNEL staining and immunoblotting analysis.

**Table 1. The primer sequences specific to each gene.**

| KLF4 | GTGCCCCGACTAACGTTG |
|---|---|
| | CGTGAACTCCCGGTCTCC |
| TNF-α | CTGAACTTCGGTGATCGG |
| | CGCTTGTGTTTGCTACTACGA |
| IL-1β | AGCAGCTTTCGACAGTGAGGA |
| | TTCATCTGGACAGCCCAAGTC |
| IL-6 | ACAAGTCCGGAGGAGACT |
| | TTGCATTGCACAACTCTTTC |
| β-actin | GCCTTCTCTCGGGTATGG |
| | AATGCCTGGGTACATGGTGG |

**Table 2. The primary antibodies employed in this study.**

| Bax | #ab32503 | Abcam |
|---|---|---|
| Bcl-2 | #ab196495 | |
| caspase-3 | #9664 | CST |
| TNF-α | #3707 | |
| IL-1β | #31202 | |
| IL-6 | #12912 | |
| β-actin | #3700 | |

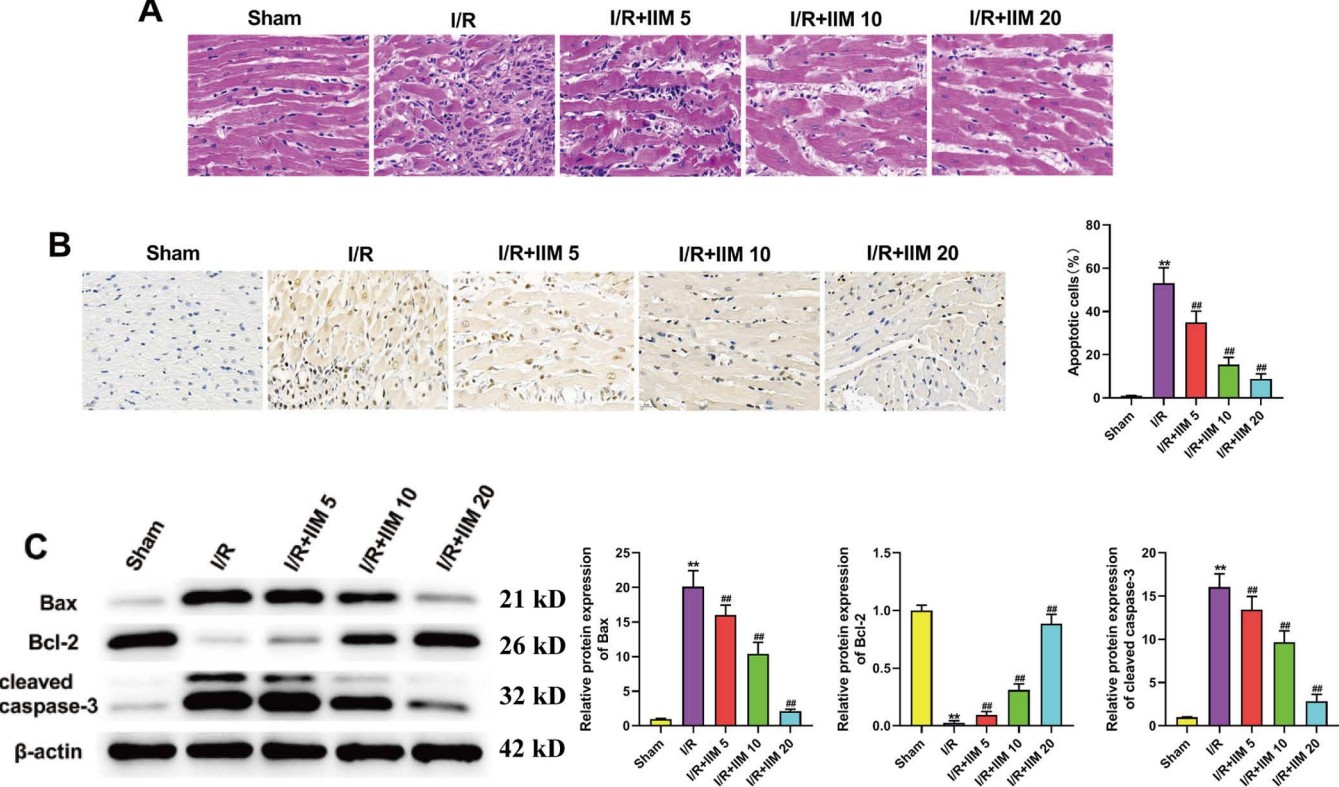

**Fig 1. IIM prevented I/R-caused MI and repressed cardiomyocyte apoptosis in rats.** (A) Representative pictures of HE staining of left ventricular heart tissue samples (5μm thick) in indicated groups (n = 5). (B) TUNEL results of myocardium in indicated groups (n = 5). (C) Representative picture of western blot and density analysis for Bax, Bcl-2 and cleaved caspase-3 in indicated groupss (n = 5). Full-length blots/gels are presented in supplementary Fig 1C. $^{**}p < 0.01$ vs. Sham; $^{#}p < 0.05$, $^{##}p < 0.01$ vs. I/R.

Compared to the Sham group, the number of TUNEL-positive cells significantly increased in the I/R group, while this phenomenon was abolished in the group treated with IIM therapy (Fig 1B). Additionally, following myocardial infarction, the expression of caspase 3 and Bax, key regulators of apoptosis, were significantly upregulated, while the expression of Bcl-2 was markedly decreased (Fig 1C). However, this effect was reversed after treatment with IIM therapy (Fig 1C).

## 2. IIM administration relieved I/R-caused inflammation response and autophagy in rats

We performed HE staining to reveal the inflammation response in rats the I/R group, (Fig S1C). Whilst, qRT-PCR results (Fig 2A) and western blot (Fig 2B) confirmed that the levels of pro-inflammatory factors (TNF-α, IL-1β and IL-6) were remarkedly upregulated in the I/R group relative to the Sham group. The upregulation of pro-inflammatory factors was significantly reversed by IIM treatment. We also investigated the role of IIM on autophagic indicators, such as Beclin 1 and LC3. The data in Fig 2C showed that I/R markedly upregulated the expression of Beclin 1 and LC3-II/LC3-I, whereas these higher expression levels were abolished by IIM. Of note, we confirmed these results through immunohistochemistry (Fig 2D).

## 3. IIM repressed apoptosis and regulated MMP in H/R-induced H9C2 cells

The results of CCK-8 assay manifested that the administration of IIM (0.25, 0.5 and 1 μM) dramatically reversed the H/R-induced decrease in H9C2 cell viability (Fig 3A). Moreover, IIM markedly reversed the upregulation of LDH levels

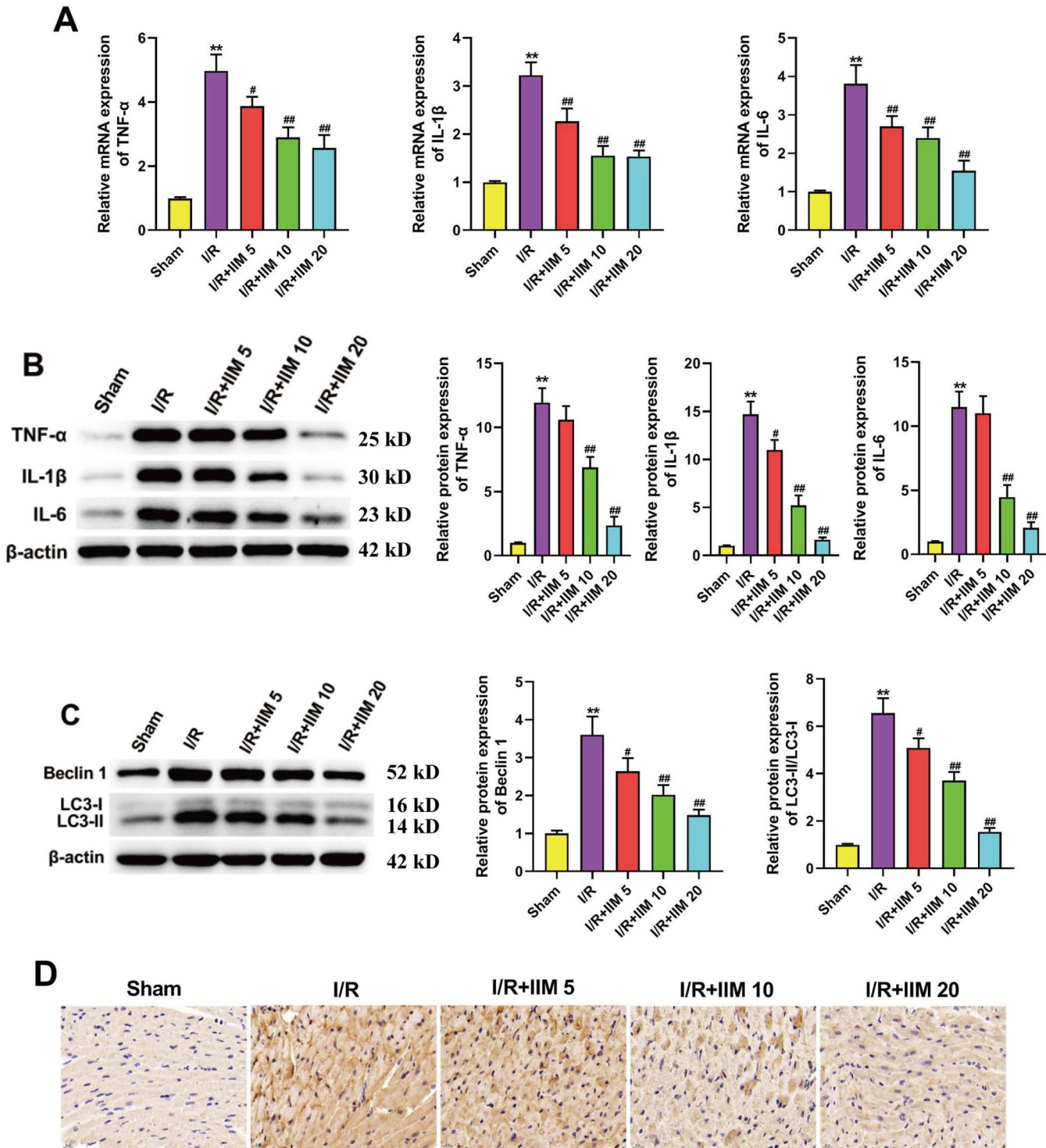

**Fig 2. IIM administration relieved I/R-induced inflammation and autophagy in rats. (A)** qRT-PCR analyses of pro-inflammatory factorsin rat heart tissues of each groups (n = 5). **(B)** Representative picture of western blot and density analysis for pro-inflammatory factorsin rat heart tissues of each groups (n = 5). **(C)** Representative picture of western blot and density analysis for Beclin 1 and LC3 in rat heart tissues of each groups (n = 5). Full-length blots/gels are presented in supplementary Fig 2B and supplementary Fig 2C. **(D)** Immunohistochemistry results of LC3 in indicated groupss (n = 5). **p < 0.01 vs. Sham; #p < 0.05, ##p < 0.01 vs. I/R.

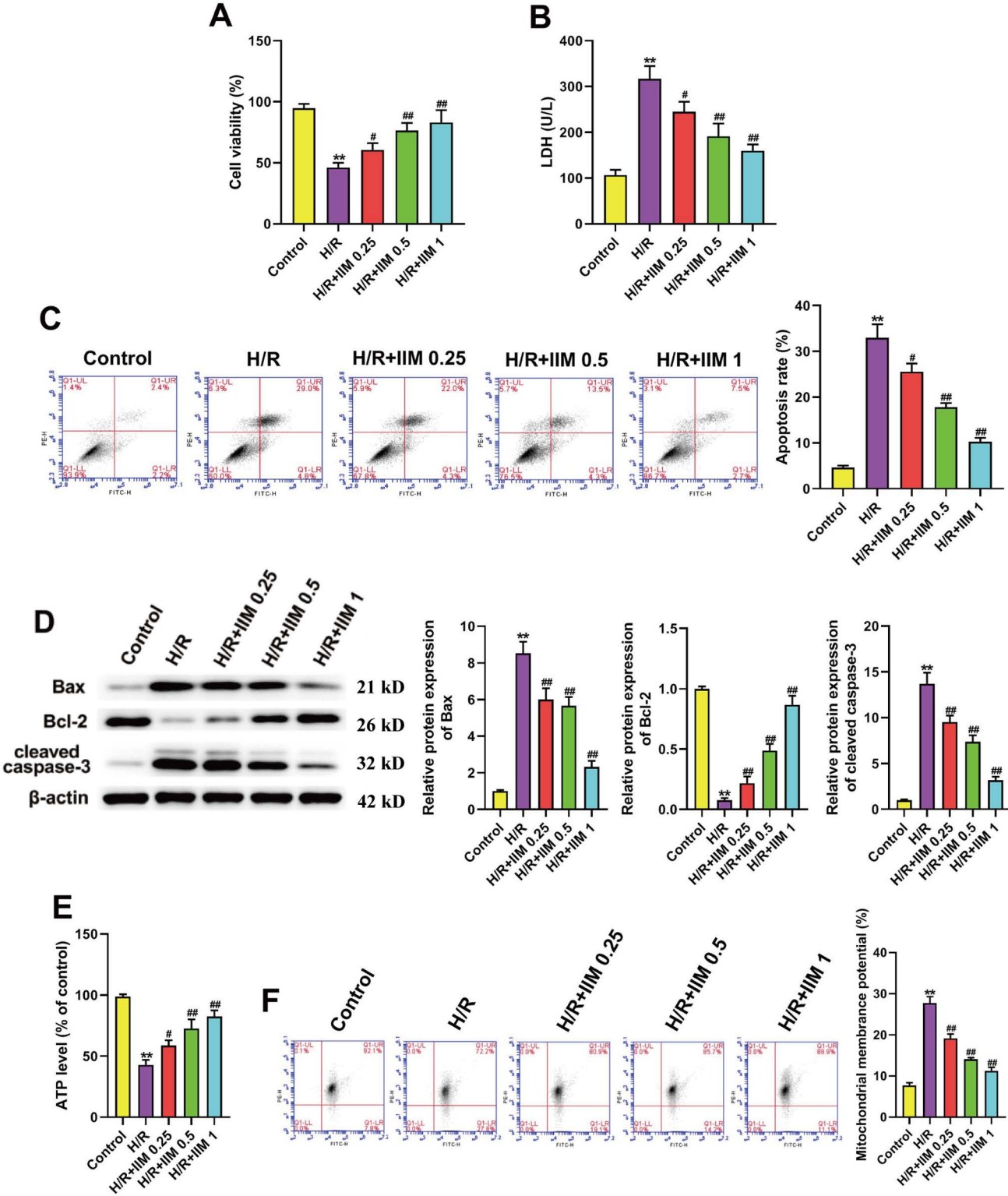

**Fig 3. IIM repressed apoptosis and regulated MMP in H/R-caused H9C2 cells.** (A) The viability of H9C2 cells in each group was tested applying CCK-8 assay. (B) LDH level of H9C2 cells were detected using LDH Cytotoxicity in indicated groups (n = 5). (C) Apoptosis in indicated groups (n = 5) was measured with a flow cytometer. (D) Representative picture of western blot and density analysis for Bax, Bcl-2 and cleaved caspase-3 in H9C2 cells of

each group (n = 5). Full-length blots/gels are presented in supplementary Fig 3D. (E) ATP level of H9C2 cells in each group were detected using ATP Assay Kits (n = 5). (F) MMP in indicated groups was detected with a flow cytometer. $^{**}p < 0.01$ vs. Control; $^{#}p < 0.05$, $^{##}p < 0.01$ vs. H/R.

(Fig 3B). As seen in Fig 3C, H9C2 cells apoptosis was dramatically elevated in the H/R group. After treatment with IIM, H9C2 cells apoptosis was significantly reduced compared to the amount of apoptosis in the H/R group. In addition, cleaved caspase 3 and Bax were highly expressed after MI, while the Bcl-2 level was markedly decreased (Fig 3D). This effect was reversed by IIM treatment (Fig 3D). Furthermore, IIM significantly reversed the H/R-induced downregulation of ATP levels (Fig 3E). Breakdown of MMP is a signature step in the progression of apoptosis. MMP was determined to elucidate IIM function on mitochondrial function. As Fig 3F shows, a higher MMP was found in the H/R group relative to the Control group. Furthermore, MMP was markedly lower after the treatment of IIM, indicating that IIM could reduce the abnormal MMP.

### 4. IIM relieved inflammation and autophagy in H/R-caused H9C2 cells

First, we explored the IIM effect on the autophagic indicator and found that Beclin 1 and LC3-II/LC3-I expression was notably elevated in the H/R group compared with the Control group (Fig 4A). Meanwhile, IIM reversed the increased expressions of Beclin 1 and LC3-II/LC3-I (Fig 4A). The immunofluorescence results further verified this phenomenon (Fig 4B). Subsequently, qRT-PCR data confirmed that H/R markedly increased pro-inflammatory factors expression, and IIM treatmentdramatically reversed the upregulation of TNF-α, IL-1β and IL-6 (Fig 4C). All these results indicated that IIM could relieve inflammation and autophagy in H/R-induced H9C2 cells.

### 5. IIM regulated the KLF4/NF-κB signaling in H/R-caused H9C2 cells

The results of qRT-PCR manifested that KLF4 was highly expressed in H/R-induced H9C2 cells (Fig 5A). Following IIM treatment, KLF4 mRNA level was significantly decreased (Fig 5B). In addition, western blot data further confirmed that H/R markedly caused the upregulation of KLF4 and p-NF-κB/NF-κB, but this was abolished after treatment with IIM. We assessed KLF4 levels after transfection with pcDNA3.1-KLF4.As seen in Fig 5C and 5D; the expression of KLF4 in the H/R + IIM 1 + KLF4 group was higher than that in the H/R + IIM 1 group. Additionally, western blot data also manifested that the ratio of p-NF-κB/NF-κB was elevated in the H/R + IIM 1 + KLF4 group compared with the H/R + IIM 1 group (Fig 5C and 5D). The expression of KLF4 in rats decreased similarly(Fig S1B). All data demonstrated that IIM treatment inhibited KLF4 and further suppressed its downstream factor, NF-κB.

### 6. KLF4 overexpression reversed the function of IIM on apoptosis, inflammation and autophagy in H/R-caused H9C2 cells

As seen in Fig 6A, the inhibitory effect of IIM on apoptosis was significantly abolished by KLF4 overexpression in H/R-induced H9C2 cells. Further, IIM function at the ATP level was assessed as an indicator of mitochondrial activity. When compared with the H/R + IIM 1 group, the ATP level was markedly reduced in the H/R + IIM 1 + KLF4 group (Fig 6B). Subsequently, we further demonstrated that the overexpression of KLF4 significantly eliminated the inhibitory role of IIM on MMP in H/ R-induced H9C2 (Fig 6C). In addition, we also confirmed that the Beclin 1 level and LC3-II/LC3-I ratio in the H/R + IIM 1 + KLF4 group was higher than that in the H/R + IIM 1 group (Fig 6D). Furthermore, the data of qRT-PCR showed that the inhibitory effect of IIM on pro-inflammatory factors was significantly reversed by KLF4 overexpression (Fig 6E).

## Discussion

Myocardial infarction (MI), caused by coronary artery disease, is a significant global cause of mortality and a growing public health concern. Unfortunately, there is currently a lack of effective approaches to improve treatment outcomes

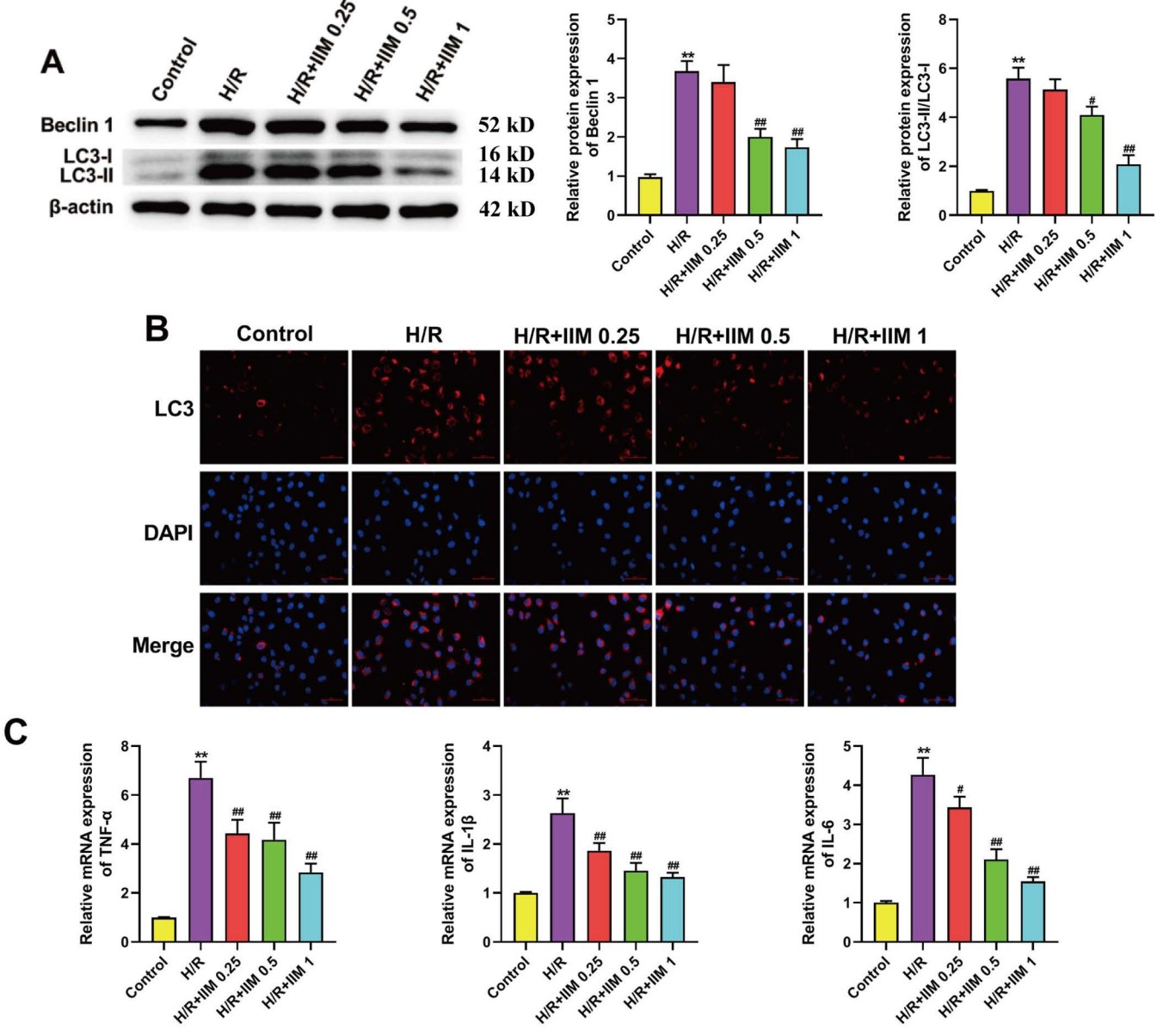

**Fig 4. IIM relieved inflammation and autophagy in H/R-induced H9C2 cells.** (A) Representative picture of western blot and density analysis for Beclin 1 and LC3 in H9C2 cells of each group (n = 5). Full-length blots/gels are presented in supplementary Fig 4A. (B) Immunofluorescence results of H9C2 cells in each group (n = 5). (C) qRT-PCR analyses of pro-inflammatory factorsin each group (n = 5). $^{**}p < 0.01$ vs. Control; $^{#}p < 0.05$, $^{##}p < 0.01$ vs. H/R.

for this condition. Given this challenging situation, our research is focused on investigating pharmacological treatment strategies for MI and understanding their underlying mechanisms. In our groundbreaking research, we have successfully demonstrated, for the first time, the effectiveness of IIM treatment in the context of MI. Additionally, we have found that the therapeutic effectiveness of this intervention directly correlates with the dosage administered. Our extensive in vivo and in vitro experiments support these conclusions, revealing the impressive ability of IIM to reduce cardiomyocyte apoptosis, inflammatory response, and autophagy processes by blocking the KLF4/NF-κB pathway.

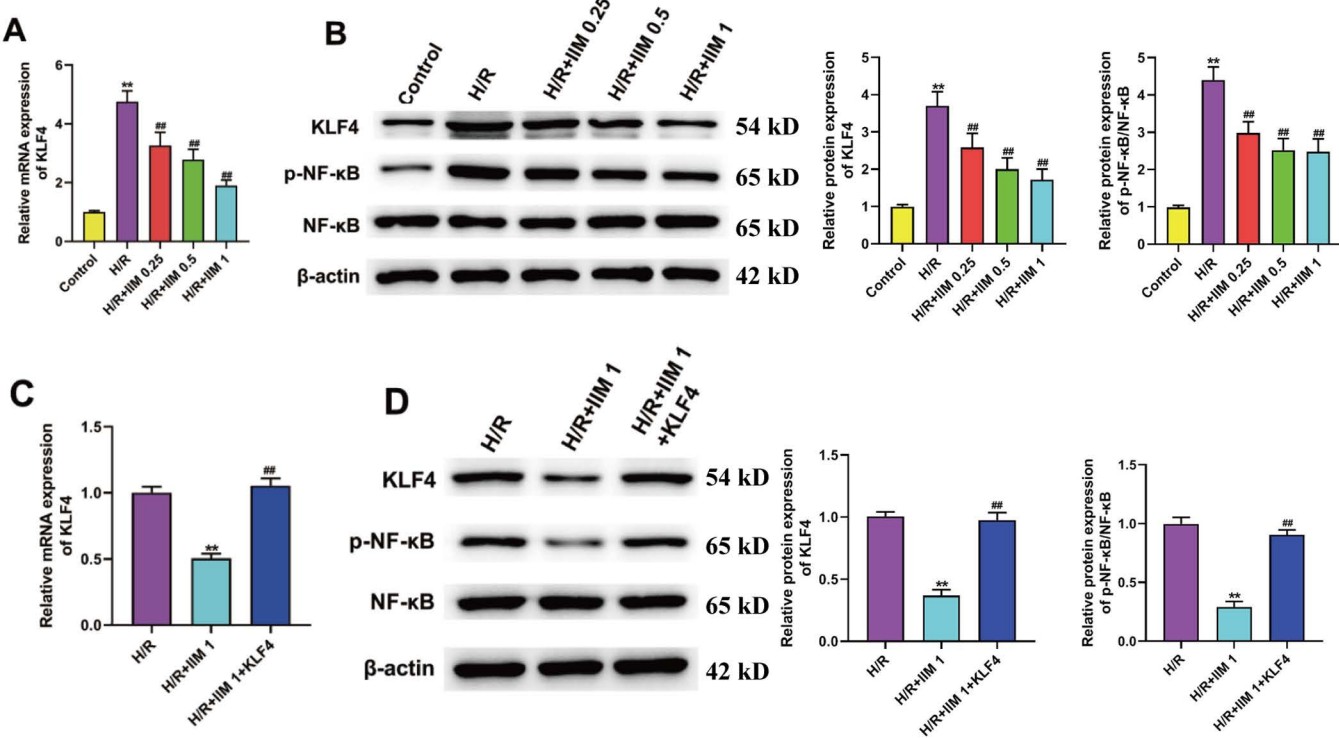

**Fig 5. IIM inhibited the KLF4/NF- κB signaling in H/R-caused H9C2 cells.** (**A**) qRT-PCR analyses of KLF4 in indicated groups (n = 5). (B) Representative picture of western blot and density analysis for KLF4, p-NF-κB and NF-κB indicated groups (n = 5). Full-length blots/gels are presented in supplementary Fig 5B. (C) After transfection of pcDNA3.1-KLF4, KLF4 level in indicated groups (n = 5) was analyzed by qRT-PCR. (D) After transfection of pcDNA3.1-KLF4, the level of KLF4, p-NF-κB and NF-κB in indicated groups (n = 5) was analyzed by western blot. Full-length blots/gels are presented in supplementary Fig 5D. $^{**}p < 0.01$ vs. Control or H/R. $^{##}p < 0.01$ vs. H/R or H/R + IIM 1.

The main mechanisms of myocardial I/R damage include inflammatory response, apoptosis, autophagy and mitochondrial damage [23–25]. Enhancing myocardial I/R damage can be achieved by improving the crucial aspect of mitochondrial energy metabolism, as stated by Tian et al.[26]. MMP serves as a marker of mitochondrial metabolic activity [27]. Mitochondria, the primary producers of adenosine triphosphate (ATP) within cells, play a pivotal role in the regulation of cellular apoptosis [28]. Previous research has shown that apoptosis hinders heart function recovery in myocardial I/R damage [29]. In this study, we observed a significant reversal of increased cardiomyocyte apoptosis induced by I/R in rats and by H/R in H9C2 cells after the administration of IIM treatment. Additionally, we have confirmed that IIM treatment effectively ameliorates abnormal MMP in H/R-induced H9C2 cells, thereby protecting against myocardial cell injury. Autophagy, a lysosomal degradation pathway, exerts important roles in maintaining cellular homeostasis and the circulation of energy and substances [30]. Several studies have suggested that excessive autophagy aggravates cardiomyocyte damage and induces apoptosis [31–33]. Our findings reveal a significant upregulation of autophagy in a rat model of myocardial I/R injury and in H9C2 cells subjected to H/R. However, we have observed that the administration of IIM effectively mitigated this enhanced autophagic response. Furthermore, our study highlights the crucial involvement of autophagy in the pathological mechanisms underlying myocardial I/R injury [34]. During myocardial I/R injury, the release of intracellular components from apoptotic cells intensifies the infiltration of inflammatory cells and triggers an inflammatory reaction within the adjacent tissue microenvironment [35,36]. Our study provides compelling evidence for the efficacy of IIM treatment in reducing the upregulation of TNF-α, IL-1β,

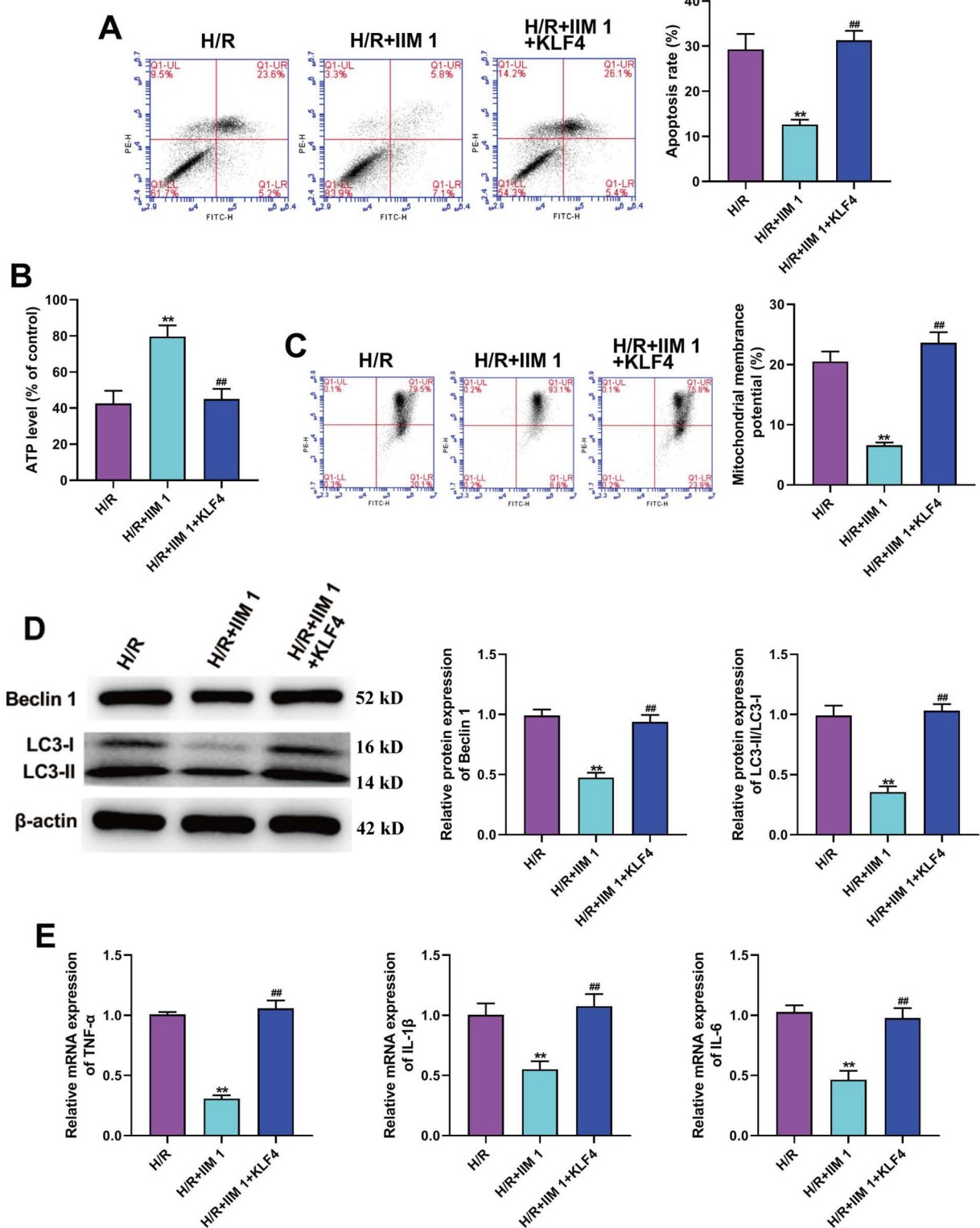

**Fig 6. KLF4 overexpression reversed the function of IIM on apoptosis, inflammation and autophagy in H/R-induced H9C2 cells. After trans-fection of pcDNA3.1-KLF4, the subsequent experiments were performed**. (A) Apoptosis in indicated groups (n = 5) was detected applying a flow

cytometer. (B) ATP level of H9C2 cells in indicated groups (n = 5). (C) MMP of H9C2 cells in indicated groups (n = 5)was tested with a flow cytometer. (D) Representative picture of western blot and density analysis for Beclin 1 and LC3 in H9C2 cells of each group (n = 5). Full-length blots/gels are presented in supplementary Fig 6D. (**E**) qRT-PCR analyses of pro-inflammatory factors in indicated groups (n = 5). **p < 0.01 vs. H/R; ##p < 0.01 vs. H/R + IIM 1.

and IL-6 in rat models of myocardial I/R injury and H9C2 cells subjected to H/R using protein imprinting and qRT-PCR techniques. These results underscore the potential of IIM as a treatment for alleviating the inflammatory reaction associated with cardiac ischemic events.

Zinc finger transcription factors, known as KLFs, are essential in a wide range of biological processes, including cell growth, programmed cell death, and cellular specialization [37]. Importantly, KLF4 has been identified as an inflammatory factor and is involved in the pro-inflammatory pathway of macrophages [18]. Furthermore, previous studies have demonstrated that upregulation of KLF4 significantly exacerbates neuroinflammation induced by amyloid-β, a major hallmark of Alzheimer's disease [38]. Conversely, inhibiting KLF4 has been shown to impede neuroinflammation caused by lipopolysaccharide (LPS) [39]. In recent years, there has been growing interest in understanding the molecular-level regulatory function of KLF4 in different chronic inflammatory reactions. Our study presents compelling evidence suggesting a significant increase in KLF4 expression in H9C2 cells subjected to H/R treatment, indicating its potential role as a pro-inflammatory mediator in myocardial infarction. Additionally, we observed an upward trend in the p-NF-κB/NF-κB ratio in H9C2 cells undergoing H/R treatment. However, treatment with IIM led to a noticeable decrease in KLF4 expression and the p-NF-κB/NF-κB ratio in H9C2 cells exposed to H/R treatment, indicating the inhibitory effect of IIM on both KLF4 expression and the downstream NF-κB pathway. We further verified a noticeable decrease in KLF4 expression after IIM treatment in rats (Fig S1B). KLF4, known as an epithelial-specific mediator of inflammation, has been shown to activate various pro-inflammatory cytokines, such as TNF-α, through the NF-κB-dependent pathway [40]. The NF-κB signaling pathway is widely recognized as a prototypical pro-inflammatory signal [41–43]. Recent evidence has revealed that KLF4/BIG1 regulated lipopolysaccharide (LPS)-induced neuroinflammation in BV2 cells via the NF-κB pathway [21].

Previous studies have elucidated the crucial role of KLF4 in the pathogenesis of sodium taurocholate-induced colitis through its regulation of the NF-κB pathway [44]. Given this, our objective was to validate the potential of KLF4 overexpression in activating the NF-κB pathway in H9C2 cells subjected to H/R insult. Additionally, we conducted a comprehensive array of rescue experiments to further substantiate the hypothesis that intervention with an IIM treatment inhibits KLF4 expression and subsequently suppresses the downstream NF-κB pathway. Thus, our findings provide compelling evidence that KLF4 overexpression counteracts H/R-induced cellular apoptosis, MMP imbalance, inflammation, and autophagy in H9C2 cells.

## Conclusions

Our investigation demonstrates the potential of IIM in improving myocardial cell apoptosis, inflammation, and autophagy triggered by myocardial infarction through the inhibition of the KLF4/NF-κB pathway in both intracellular and extracellular environments. This groundbreaking finding not only enhances our understanding of the underlying mechanisms of myocardial infarction but also paves the way for the development of more effective therapeutic strategies. However, it is important to note that our study did not investigate the in vivo expression of KLF4, which represents a limitation of our research. Future investigations should incorporate animal models or clinical samples to assess the expression of KLF4 in the context of myocardial ischemia-reperfusion injury. On the other hand, because Iron plays a critical role in ROS generation, the levels of oxidative stress injury and the autophagic death will be detected. This will provide deeper insights into the precise mechanism underlying the inhibitory effect of ischemic postconditioning on the KLF4/NF-κB pathway and explore additional mechanisms associated with myocardial infarction.

## Supporting information

**Fig S1. IIM administration relieved I/R-induced inflammation and myocardial infarction in rats.** IIM supplementation decreased ferritin concentration in the IR group and IR+IIM group (p<0.05, n=3). Representative picture of density analysis for KLF4 in rat heart tissues of each group (n=3). Full-length blots/gels are presented in supplementary for Fig S1. **(C)** TTC results of myocardial infarction in rats in indicated groups. 1–3 were the IR group, 4–6 were the IR+IIM group. The myocardial slices showed clear infarct areas with infarct size attached. **(D)** HE staining for inflammatory infiltration of myocardial cells.
(PDF)

## Author contributions

**Data curation:** Huiping Gong, Duanchen Sun.

**Formal analysis:** Qingyang Zhao.

**Funding acquisition:** Huiping Gong, Qiaofeng Dong, Aixia Dou.

**Investigation:** Huiping Gong.

**Methodology:** Jingbo Zhang, Qiaofeng Dong.

**Project administration:** Qingyang Zhao, Jingbo Zhang, Xianghua Zhuang.

**Resources:** Qiaofeng Dong.

**Software:** Duanchen Sun.

**Supervision:** Duanchen Sun, Xianghua Zhuang, Aixia Dou.

**Validation:** Qingyang Zhao, Xianghua Zhuang.

**Writing – review & editing:** Aixia Dou.

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
