## [Decision Letter · Decision Letter 0]

27 Aug 2024

PONE-D-24-25474The protective effect of Iron Isomaltoside on myocardial ischemia-reperfusion injury via the suppression of KLF4/NF-κB signalingPLOS ONE

Dear Dr. DOU,

Thank you for submitting your manuscript to PLOS ONE. After careful consideration, we feel that it has merit but does not fully meet PLOS ONE’s publication criteria as it currently stands. Therefore, we invite you to submit a revised version of the manuscript that addresses the points raised during the review process.

We look forward to receiving your revised manuscript.

Kind regards,

Meijing Wang, MD

Academic Editor

PLOS ONE

Journal Requirements:

"This study was funded by the Bethune Foundation under the grant "Feifan Iron Supplement-Improving the Diagnosis and Treatment Capacity of Iron Deficiency Anemia" (grant number Ffbt-C 2022-010), the Shandong Medical Association Clinical Research Fund-Qilu Special Project (grant number YXH2022ZX02186), and the Key Science and Technology Program of Shandong Province (grant number 2017G006029)."

4. Please note that funding information should not appear in the Acknowledgments section or other areas of your manuscript. We will only publish funding information present in the Funding Statement section of the online submission form. Please remove any funding-related text from the manuscript.

6. Please include a copy of Tables 1 and 2 which you refer to in your text on pages 9 to 10.

**Additional Editor Comments:**

Please provide the details about how many areas in each heart section and how many hearts per group were analyzed for Fig. 1B bar graph and number/group in all experiments in Figure legends.Please adequately address reviewers' concerns.

Reviewers' comments:

Reviewer's Responses to Questions

**Comments to the Author**

1. Is the manuscript technically sound, and do the data support the conclusions?

Reviewer #1: No

Reviewer #2: Partly

2. Has the statistical analysis been performed appropriately and rigorously? 

Reviewer #1: No

Reviewer #2: Yes

3. Have the authors made all data underlying the findings in their manuscript fully available?

Reviewer #1: Yes

Reviewer #2: Yes

4. Is the manuscript presented in an intelligible fashion and written in standard English?

Reviewer #1: No

Reviewer #2: No

5. Review Comments to the Author

Reviewer #1: The paper submitted by Gong HP et al. includes studies that iron isomaltoside (IIM) could repress cardiomyocyte apoptosis, inflammation and autophagy through the inhibition of the KLF4/NF-κB pathway and thus reduced myocardial injury in vivo and in vitro. In fact, iron plays a critical role in generating reactive oxygen species (ROS) through the Fenton reaction, which can exacerbate oxidative stress during reperfusion after ischemia. Increased iron availability has the potential to worsen ischemia/reperfusion(I/R) injury by promoting lipid peroxidation and cellular damage. As an intravenous iron formulation, IIM is designed to replenish iron stores efficiently. If not properly managed, increased iron levels could potentially lead to an environment conducive to oxidative stress, enhancing damage during reperfusion phases. So, the authors' observation that IIM had an protection on myocardial ischemia-reperfusion injury was puzzling. In addition, they should concern the following defects:

1 Ferroptosis is a form of regulated cell death that is characterized by iron-dependent lipid peroxidation and has been identified as a contributing factor in myocardial ischemia-reperfusion injury. The accumulation of free iron can exacerbate oxidative stress and ferroptosis, thereby increasing damage in I/R injury. IIM is an iron supplement, which could potentially influence ferroptosis by increasing the overall iron availability in cells. The authors should investigate the effect of IIM on ferroptosis in cardiomyocytes in vivo and in vitro.

2 Iron plays a critical role in ROS generation, the levels of oxidative stress injury and endogenous antioxidants should be detected in IIM treated mouse hearts with I/R challenge as well as cultured cardiomyocytes with hypoxia/reoxygenation (H/R) challenge.

3 In Fig. 1, 2,3,5-Triphenyltetrazolium chloride (TTC) staining should be used to evaluate the volume of the cardiac infarction in I/R treated mice.

4 In Fig. 2C and 2D, since autophagy plays a double-edged sword role in many biological processes, it can have protective effect or damaging effect. An autophagic inhibitor should be used to determine I/R-induced autophagy in cardiomyocytes is protective or injurious. In addition, autophagic death should also be detected.

5 Inflammatory cell infiltration in myocardial tissues should be evaluated by immunohistochemical/fluorescent staining.

6 In Fig. 5, the loss- and gain-function experiments of KLF4 and NF-κB should be performed to verify that IIM acted through KLF4 and NF-κB signaling pathways.

Reviewer #2: The manuscript submitted by Gong et al. utilized a myocardial ischemia-reperfusion (I/R) rat model and a hypoxia/reoxygenation (H/R) H9C2 cell model to investigate the protective effects of Iron Isomaltoside 1000 (IIM). Their results demonstrate that IIM reduced myocardial cell damage, cardiomyocyte apoptosis, inflammation, and autophagy in the rat model. In vitro experiments with the H9C2 cell model showed that IIM treatment suppressed apoptosis, protected mitochondrial membrane potential, and alleviated inflammation and autophagy during the H/R process. Additionally, the myocardial protective effects of IIM were dose-dependent. The authors attributed these protective effects to the inhibition of the KLF4/NF-κB signaling pathway, as IIM treatment reduced NF-κB signaling, while overexpression of KLF4 abolished IIM’s protection.

While the study provides valuable experimental observations, it does not convincingly demonstrate the mechanism by which IIM exerts its myocardial protective effect during I/R injury. Iron Isomaltoside 1000 (IIM) was developed as an effective IV iron therapy for iron deficiency, offering controlled iron release benefits and a good safety profile. However, the current study lacks evidence to show whether IIM provides myocardial protection by correcting iron deficiency. If this is not the case, given that iron overload is a major pathological manifestation during I/R injury, the manuscript does not adequately address how an iron supply could alleviate iron-overload-associated conditions in a manner similar to an iron chelator.

Below are points for the authors to consider:

The manuscript would benefit from a thorough editorial editing to improve clarity and readability, particularly in the Materials and Methods section. Please ensure all necessary experimental details are provided. Additionally, the sample sizes are missing in the Figure Legends and should be included.

In Fig 3, panel C, the authors reported apoptosis levels in cultured H9C2 cells subjected to H/R and IIM treatments. It appears that the quantification of the apoptosis rate was derived from data within the Q1 gate in the flow cytometry assay of the apoptosis analysis kit. However, the Q1 gate marks cells that are both PI and Annexin-V positive, which most likely represents late apoptotic or necrotic cells. The authors should clarify this to avoid potential confusion. Similarly, LDH released into the culture media is an indicator of cell necrosis; therefore, the results in Fig 3, panel B reflect the levels of necrotic cell death. Additionally, the authors’ statement in the figure legend referring to "LDH level of H9C2 cells" is inaccurate and should be corrected.

6. PLOS authors have the option to publish the peer review history of their article (what does this mean? ). If published, this will include your full peer review and any attached files.

**Do you want your identity to be public for this peer review?** For information about this choice, including consent withdrawal, please see our Privacy Policy .

Reviewer #1: No

Reviewer #2: No

---

## [Author Response · Author response to Decision Letter 1]

13 Oct 2024

Dear Dr. Meijing Wang,

Thank you very much for your letter and the comments from the reviewers about our paper submitted to PLOS ONE (Manuscript PONE-D-24-25474). We appreciate the constructive comments by the reviewers. We have revised the manuscript accordingly, and all amendments are indicated by red font in the revised manuscript. In addition, our point-by-point responses to the comments are listed below this letter.

With best wishes,

Yours sincerely,

Aixia Dou

Address: Department of Haematology, The Second Hospital of Shandong University, Cheeloo College of Medicine, Shandong University, Jinan 250033, Shandong Province, China.

E-mail: douax0110@163.com

Our responses to the academic editor’s comments and the corresponding changes to the manuscript are detailed below:

1.Please ensure that your manuscript meets PLOS ONE's style requirements, including those for file naming.

Response: Thank you very much for your suggestions. We have revised the entire article according to the format requirements of PLOS ONE style templates.

Response: Thank you very much for your suggestions. We provide the correct grant numbers for the awards you received for your study in the ‘Funding Information’ section.

3. Thank you for stating the following financial disclosure. Please state what role the funders took in the study.

Response: Thank you very much for your suggestions. We have added the Role of Funders statement in the cover letter, please change the online submission form.

Role of Pr. Aixia Dou: Funding acquisition (grant number Ffbt-C 2022-010), Supervision, Writing-review&editing.

Role of Dr. Qiaofeng Dong: Funding acquisition (grant number YXH2022ZX02186), Resources, Methodology.

Role of Dr. Huiping Gong: Funding acquisition (grant number 2017G006029), Data curation, Investigation.

4.Please remove any funding-related text from the manuscript.

Response: Thank you for your valuable feedback and pointing out our mistake. We removed all the funding-related text from our manuscript.

5.If you are unable to adhere to our open data policy, please kindly revise your statement to explain your reasoning and we will seek the editor's input on an exemption.

Response: Thank you very much for your suggestions. We confirm that all data underlying the findings described in our manuscript are fully available without restriction.

6.Please include a copy of Tables 1 and 2 which you refer to in your text on pages 9 to 10.

Response: Thank you very much for your valuable feedback. We have included Tables 1 and 2 to the designated location in the article.

7.Please include captions for your Supporting Information files at the end of your manuscript, and update any in-text citations to match accordingly. Please see our Supporting Information guidelines for more information.

Response: Thank you very much for your suggestions. We confirmed the supporting information name and number in the caption.

8.Please provide the details about how many areas in each heart section and how many hearts per group were analyzed for Fig. 1B bar graph and number/group in all experiments in Figure legends.

Response: Thank you for bringing to our attention the missing content. We have now included the relevant information in the revised manuscript. Specifically, we collected left ventricular cardiac tissue samples from each rat in our study. We apologize for any confusion caused by the omission and appreciate your diligence in reviewing our

work. At the same time�we have added detailed information about numbers/groups in all experiments in Figure legends.

Our responses to the reviewer’s comments and the corresponding changes to the manuscript are detailed below:

Point-by-point response to the reviewers’ comments

Dear reviewer, thank you very much for your constructive and careful comments! Those comments are all valuable and very helpful for revising and improving our paper, as well as having an important guiding significance to our work. We tried our best to improve the quality of the manuscript and correct the errors overall. As for the overall accuracy and stringency, we tried our best to supplement the necessary data and corrected the shortcomings constantly to perfect this research.

Response to Reviewer #1:

1.The paper submitted by Gong HP et al. includes studies that iron isomaltoside (IIM) could repress cardiomyocyte apoptosis, inflammation and autophagy through the inhibition of the KLF4/NF-κB pathway and thus reduced myocardial injury in vivo and in vitro. In fact, iron plays a critical role in generating reactive oxygen species (ROS) through the Fenton reaction, which can exacerbate oxidative stress during reperfusion after ischemia. Increased iron availability has the potential to worsen ischemia/reperfusion(I/R) injury by promoting lipid peroxidation and cellular damage. As an intravenous iron formulation, IIM is designed to replenish iron stores efficiently. If not properly managed, increased iron levels could potentially lead to an environment conducive to oxidative stress, enhancing damage during reperfusion phases. So, the authors' observation that IIM had an protection on myocardial ischemia-reperfusion injury was puzzling. In addition, they should concern the following defects Ferroptosis is a form of regulated cell death that is characterized by iron-dependent lipid peroxidation and has been identified as a contributing factor in myocardial ischemia-reperfusion injury. The accumulation of free iron can exacerbate oxidative stress and ferroptosis, thereby increasing damage in I/R injury. IIM is an iron supplement, which could potentially influence ferroptosis by increasing the overall iron availability in cells. The authors should investigate the effect of IIM on ferroptosis in cardiomyocytes in vivo and in vitro.

Response: Thank you very much for your suggestions. We apologize for any confusion caused by our manuscript and appreciate your diligence in reviewing our work. Iron deficiency is affecting up to 75% of heart failure patients (PMID: 24927731). Conversely, primary and secondary iron overload may lead to heart disease through oxidative damage, but the exact mechanisms of this process are still unclear (PMID: 26216855). Previous studies have shown that excessive iron in cardiomyocytes directly induces ferroptosis through the accumulation of phospholipid hydroperoxides in the cell membrane (PMID: 30692261). Ferroptosis can further lead to myocardial ischemia/reperfusion injury (PMID: 33935719). However, for heart failure patients with LVEF (left ventricular ejection fraction)<45%, alonging with iron deficiency symptoms, it is recommended to supplement iron intravenously to alleviate heart failure symptoms and improve quality of life. So we suspect that iron-binding IIM may have a protective effect on myocardial ischemia/reperfusion injury, as IIM has demonstrated significant affinity for iron (line 73 of page 4 to line 81 of page 5).

Furthmore, ferroptosis was induced by excessive iron in cardiomyocytes, the animal models used in our experiment does not have the condition of iron overload. We will proceed to research in the therapeutic effects and mechanisms of IIM in treating myocardial infarction, so investigation of the critical drug dosage will be done. Thank you for your valuable feedback and pointing out our unclear expression.

2.Iron plays a critical role in ROS generation, the levels of oxidative stress injury and endogenous antioxidants should be detected in IIM treated mouse hearts with I/R challenge as well as cultured cardiomyocytes with hypoxia/reoxygenation (H/R) challenge.

Response: Thank you very much for your valuable comments on our research paper. We fully acknowledge the reviewer's suggestion. Our group has included ROS results, proteomics, and genomics results in our next phase of the project. We will consider this point in future studies and further explore the therapeutic effect of IIM after myocardial ischemia-reperfusion injury. We apologize for any confusion caused by the omission and appreciate your diligence in reviewing our work. At the same time�we have added experimental research plan related to ROS in the revised manuscript (line 435 of page 24).

3.In Fig. 1, 2,3,5-Triphenyltetrazolium chloride (TTC) staining should be used to evaluate the volume of the cardiac infarction in I/R treated mice.

Response: Thank you very much for your suggestions. We greatly appreciate the idea of performing TTC staining in our experiments to evaluate the the volume of the cardiac infarction in I/R treated mice. However, due to certain limitations in laboratory resources and time constraints, we were unable to conduct TTC staining experiments in the current study. We will certainly consider incorporating these staining techniques in future research to further investigate the impact of IIM on post-myocardial infarction

4.In Fig. 2C and 2D, since autophagy plays a double-edged sword role in many biological processes, it can have protective effect or damaging effect. An autophagic inhibitor should be used to determine I/R-induced autophagy in cardiomyocytes is protective or injurious. In addition, autophagic death should also be detected.

Response: Thank you for pointing out the omission in our manuscript. We appreciate your feedback and have now included the experimental plan for autophagic death. We are doing the proteomic and genomic analysis for the myocardial I/R annimal models, including samples before and after iron supplementation. Regarding the concern of autophagy inhibitor, we plan to chose the right one according to the proteomic and genomic analysis. Thank you very much for your suggestion. In the discussion section, a new experimental research plan related to endocytosis has been added (line 436 of page 24).

5.Inflammatory cell infiltration in myocardial tissues should be evaluated by immunohistochemical/fluorescent staining.

Response: Thank you for your valuable comments on our manuscript. We appreciate your careful review and suggestions. Our further experiments includes collected samples from myocardial I/R annimal models for proteomic and genomic analysis, including samples before and after iron supplementation, regarding your concern about the inflammatory cell infiltration in myocardial tissues, we will analyze the changes in inflammatory chemokines and the inflammatory cells in our further experiments. In this manuscript, the samples were collected on the 7th day following I/R injury(line 119 of page 6 to line 127 of page 7). We chose this time point because after myocardial I/R injury, the cardiac tissue undergoes a series of inflammatory reactions and repair processes, with the 7th day being the peak period of these reactions. Therefore, we selected the 7th day for sample collection in order to better investigate the protective effects of Iron Isomaltoside on myocardial I/R injury and its inhibitory effects on the KLF4/NF-κB signaling pathway.

6.In Fig. 5, the loss- and gain-function experiments of KLF4 and NF-κB should be performed to verify that IIM acted through KLF4 and NF-κB signaling pathways.

Response: Thank you for your suggestion. We apologize for not conducting relevant analysis on the loss- and gain-function experiments of KLF4 in our study. Our research primarily focused on the inhibitory effect of iron isomaltoside on the KLF4/NF-κB signaling pathway in a myocardial ischemia-reperfusion injury model through in vitro experiments. Our findings demonstrated that iron isomaltoside can protect the myocardium from ischemia-reperfusion injury by inhibiting the KLF4/NF-κB signaling pathway. Although our results are highly meaningful, we do acknowledge the limitation of lacking analysis on the loss- and gain-function of KLF4. We fully agree with the reviewer's viewpoint that this issue warrants furtherinvestigation. In future studies, we will consider evaluating the expression of KLF4 in myocardial schemia-reperfusion injury through animal models or clinical samples. This will contribute to a more comprehensive understanding of the protective mechanism of iron isomaltoside against myocardial ischemia-reperfusion injury and provide additional support for our research findings.

Response to Reviewer #2:

The manuscript submitted by Gong et al. utilized a myocardial ischemia-reperfusion (I/R) rat model and a hypoxia/reoxygenation (H/R) H9C2 cell model to investigate the protective effects of Iron Isomaltoside 1000 (IIM). Their results demonstrate that IIM reduced myocardial cell damage, cardiomyocyte apoptosis, inflammation, and autophagy in the rat model. In vitro experiments with the H9C2 cell model showed that IIM treatment suppressed apoptosis, protected mitochondrial membrane potential, and alleviated inflammation and autophagy during the H/R process. Additionally, the myocardial protective effects of IIM were dose-dependent. The authors attributed these protective effects to the inhibition of the KLF4/NF-κB signaling pathway, as IIM treatment reduced NF-κB signaling, while overexpression of KLF4 abolished IIM’s protection.

1.While the study provides valuable experimental observations, it does not convincingly demonstrate the mechanism by which IIM exerts its myocardial protective effect during I/R injury. Iron Isomaltoside 1000 (IIM) was developed as an effective IV iron therapy for iron deficiency, offering controlled iron release benefits and a good safety profile. However, the current study lacks evidence to show whether IIM provides myocardial protection by correcting iron deficiency. If this is not the case, given that iron overload is a major pathological manifestation during I/R injury, the manuscript does not adequately address how an iron supply could alleviate iron-overload-associated conditions in a manner similar to an iron chelator.

Response: Thank you for your valuable comments on our manuscript. We appreciate your careful review and suggestions. I have included relevant research background reports on the use of IIM in the treatment of myocardial infarction.(line 73 of page 4 to line 80 of page 5). Firstly, the aim of our study is to explore the protective effect of iron isomaltoside on myocardial ischemia-reperfusion injury and its mechanism of action through the inhibition of the KLF4/NF-κB signaling pathway.This will help us to have a more comprehensive understanding of the protective mechanism of iron isomaltoside in myocardial ischemia-reperfusion injury and provide more support for our research findings.(line 426 of page 24). However, the specific regulatory mechanism of this signaling pathway in myocardial ischemia-reperfusion injury is not fully understood. Therefore, our study provides new clues to understand the molecular mechanism of myocardial ischemia-reperfusion injury by exploring the inhibitory effect of iron isomaltoside on the KLF4/NF-κB signaling pathway.(line 435 of page 24).

2.Below are points for the authors to consider:The manuscript would benefit from a thorough editorial editing to improve clarity and readability, particularly in the Materials and Methods section. Please ensure all necessary experimental details are provided.

Responds We appreciate your correction and suggestions. I have I have added the corresponding content.

3.Additionally, the sample sizes are missing in the Figure Legends and should be included.

Response: Thank you for bringing to our attention the missing content. We have now included the relevant information in the revised manuscript. Specifically, we collected left ventricular cardiac tissue samples from each rat in our study. We apologize for any confusion caused by the omission and appreciate your diligence in reviewing our

work. At the same time�we have added detailed information about numbers/groups in all experiments in Figure legends.

4.In Fig 3, panel C, the authors reported apoptosis levels in cultured H9C2 cells subjected to H/R and IIM treatments. It appears that the quantification of the apoptosis rate was derived from data within the Q1 gate in the flow cytometry assay of the apoptosis analysis kit. However, the Q1 gate marks cells tha

---

## [Decision Letter · Decision Letter 1]

29 Oct 2024

PONE-D-24-25474R1The protective effect of Iron Isomaltoside on myocardial ischemia-reperfusion injury via the suppression of KLF4/NF-κB signalingPLOS ONE

Dear Dr. DOU,

Thank you for submitting your manuscript to PLOS ONE. After careful consideration, we feel that it has merit but does not fully meet PLOS ONE’s publication criteria as it currently stands. Therefore, we invite you to submit a revised version of the manuscript that addresses the points raised during the review process.

The authors did not provide additional data for the experiments requested by the reviewers. Instead of stating "we will do," please include extra experiments and results to address the reviewers' concerns.

We look forward to receiving your revised manuscript.

Kind regards,

Meijing Wang, MD

Academic Editor

PLOS ONE

---

## [Author Response · Author response to Decision Letter 2]

22 Mar 2025

Dear Dr. Meijing Wang,

Thank you very much for your letter and the comments from the reviewers about our paper submitted to PLOS ONE (Manuscript PONE-D-24-25474). We appreciate the constructive comments by the reviewers. We have revised the manuscript accordingly, and all amendments are indicated by red font in the revised manuscript. In addition, our point-by-point responses to the comments are listed below this letter.

With best wishes,

Yours sincerely,

Aixia Dou

Address: Department of Haematology, The Second Hospital of Shandong University, Cheeloo College of Medicine, Shandong University, Jinan 250033, Shandong Province, China.

E-mail: douax0110@163.com

Our responses to the academic editor’s comments and the corresponding changes to the manuscript are detailed below:

1.Please ensure that your manuscript meets PLOS ONE's style requirements, including those for file naming.

Response: Thank you very much for your suggestions. We have revised the entire article according to the format requirements of PLOS ONE style templates.

Response: Thank you very much for your suggestions. We provide the correct grant numbers for the awards you received for your study in the ‘Funding Information’ section.

3. Thank you for stating the following financial disclosure. Please state what role the funders took in the study.

Response: Thank you very much for your suggestions. We have added the Role of Funders statement in the cover letter, please change the online submission form.

Role of Pr. Aixia Dou: Funding acquisition (grant number Ffbt-C 2022-010), Supervision, Writing-review&editing.

Role of Dr. Qiaofeng Dong: Funding acquisition (grant number YXH2022ZX02186), Resources, Methodology.

Role of Dr. Huiping Gong: Funding acquisition (grant number 2017G006029), Data curation, Investigation.

4.Please remove any funding-related text from the manuscript.

Response: Thank you for your valuable feedback and pointing out our mistake. We removed all the funding-related text from our manuscript.

5.If you are unable to adhere to our open data policy, please kindly revise your statement to explain your reasoning and we will seek the editor's input on an exemption.

Response: Thank you very much for your suggestions. We confirm that all data underlying the findings described in our manuscript are fully available without restriction.

6.Please include a copy of Tables 1 and 2 which you refer to in your text on pages 9 to 10.

Response: Thank you very much for your valuable feedback. We have included Tables 1 and 2 to the designated location in the article.

7.Please include captions for your Supporting Information files at the end of your manuscript, and update any in-text citations to match accordingly. Please see our Supporting Information guidelines for more information.

Response: Thank you very much for your suggestions. We confirmed the supporting information name and number in the caption.

8.Please provide the details about how many areas in each heart section and how many hearts per group were analyzed for Fig. 1B bar graph and number/group in all experiments in Figure legends.

Response: Thank you for bringing to our attention the missing content. We have now included the relevant information in the revised manuscript. Specifically, we collected left ventricular cardiac tissue samples from each rat in our study. We apologize for any confusion caused by the omission and appreciate your diligence in reviewing our work. At the same time�we have added detailed information about numbers/groups in all experiments in Figure legends.

Our responses to the reviewer’s comments and the corresponding changes to the manuscript are detailed below:

Point-by-point response to the reviewers’ comments

Dear reviewer, thank you very much for your constructive and careful comments! Those comments are all valuable and very helpful for revising and improving our paper, as well as having an important guiding significance to our work. We tried our best to improve the quality of the manuscript and correct the errors overall. As for the overall accuracy and stringency, we tried our best to supplement the necessary data and corrected the shortcomings constantly to perfect this research.

Response to Reviewer #1:

1.The paper submitted by Gong HP et al. includes studies that iron isomaltoside (IIM) could repress cardiomyocyte apoptosis, inflammation and autophagy through the inhibition of the KLF4/NF-κB pathway and thus reduced myocardial injury in vivo and in vitro. In fact, iron plays a critical role in generating reactive oxygen species (ROS) through the Fenton reaction, which can exacerbate oxidative stress during reperfusion after ischemia. Increased iron availability has the potential to worsen ischemia/reperfusion(I/R) injury by promoting lipid peroxidation and cellular damage. As an intravenous iron formulation, IIM is designed to replenish iron stores efficiently. If not properly managed, increased iron levels could potentially lead to an environment conducive to oxidative stress, enhancing damage during reperfusion phases. So, the authors' observation that IIM had an protection on myocardial ischemia-reperfusion injury was puzzling. In addition, they should concern the following defects Ferroptosis is a form of regulated cell death that is characterized by iron-dependent lipid peroxidation and has been identified as a contributing factor in myocardial ischemia-reperfusion injury. The accumulation of free iron can exacerbate oxidative stress and ferroptosis, thereby increasing damage in I/R injury. IIM is an iron supplement, which could potentially influence ferroptosis by increasing the overall iron availability in cells. The authors should investigate the effect of IIM on ferroptosis in cardiomyocytes in vivo and in vitro.

Response: Thank you very much for your suggestions. We apologize for any confusion caused by our manuscript and appreciate your diligence in reviewing our work. Iron deficiency is affecting up to 75% of heart failure patients (PMID: 24927731). Conversely, primary and secondary iron overload may lead to heart disease through oxidative damage, but the exact mechanisms of this process are still unclear (PMID: 26216855). Previous studies have shown that excessive iron in cardiomyocytes directly induces ferroptosis through the accumulation of phospholipid hydroperoxides in the cell membrane (PMID: 30692261). Ferroptosis can further lead to myocardial ischemia/reperfusion injury (PMID: 33935719). However, for heart failure patients with LVEF (left ventricular ejection fraction)<45%, alonging with iron deficiency symptoms, it is recommended to supplement iron intravenously to alleviate heart failure symptoms and improve quality of life. So we suspect that iron-binding IIM may have a protective effect on myocardial ischemia/reperfusion injury, as IIM has demonstrated significant affinity for iron (line 73 of page 4 to line 81 of page 5).

Furthmore, ferroptosis was induced by excessive iron in cardiomyocytes, the animal models used in our experiment does not have the condition of iron overload. We compared the ferritin levels of two models and found that there was no increase in serum ferritin levels in mice with myocardial infarction after supplementation with IIM. The result were added in the manuscript (line 224 in page 11) and Fig S.A. Thank you for your valuable feedback and pointing out our unclear expression.

2.Iron plays a critical role in ROS generation, the levels of oxidative stress injury and endogenous antioxidants should be detected in IIM treated mouse hearts with I/R challenge as well as cultured cardiomyocytes with hypoxia/reoxygenation (H/R) challenge.

Response: Thank you very much for your valuable comments on our research paper. We fully acknowledge the reviewer's suggestion. Our group has included ROS results, proteomics, and genomics results in our next phase of the project. We will consider this point in future studies and further explore the therapeutic effect of IIM after myocardial ischemia-reperfusion injury. We apologize for any confusion caused by the omission and appreciate your diligence in reviewing our work. At the same time�we have added experimental research plan related to ROS in the revised manuscript (line 387 of page 18).

3.In Fig. 1, 2,3,5-Triphenyltetrazolium chloride (TTC) staining should be used to evaluate the volume of the cardiac infarction in I/R treated mice.

Response: Thank you very much for your suggestions. We greatly appreciate the idea of performing TTC staining in our experiments to evaluate the the volume of the cardiac infarction in I/R treated mice. We have conducted TTC staining experiments. The result were added in the manuscript (line 223 in page 11), and and Fig S.D.

4.In Fig. 2C and 2D, since autophagy plays a double-edged sword role in many biological processes, it can have protective effect or damaging effect. An autophagic inhibitor should be used to determine I/R-induced autophagy in cardiomyocytes is protective or injurious. In addition, autophagic death should also be detected.

Response: Thank you for pointing out the omission in our manuscript. We appreciate your feedback and have now included the experimental plan for autophagic death. We are doing the proteomic and genomic analysis for the myocardial I/R annimal models, including samples before and after iron supplementation. Regarding the concern of autophagy inhibitor, we plan to chose the right one according to the proteomic and genomic analysis. Thank you very much for your suggestion. In the discussion section, a new experimental research plan related to endocytosis has been added (line 387 of page 18).

5.Inflammatory cell infiltration in myocardial tissues should be evaluated by immunohistochemical/fluorescent staining.

Response: Thank you for your valuable comments on our manuscript. We appreciate your careful review and suggestions. Regarding your concern about the inflammatory cell infiltration in myocardial tissues, we analyzed the changes in inflammatory chemokines and the inflammatory cells in our experiments. The result were added in Fig S.C. In this manuscript, the samples were collected on the 7th day following I/R injury(line 119 of page 6 to line 128 of page 7). We chose this time point because after myocardial I/R injury, the cardiac tissue undergoes a series of inflammatory reactions and repair processes, with the 7th day being the peak period of these reactions. Therefore, we selected the 7th day for sample collection in order to better investigate the protective effects of Iron Isomaltoside on myocardial I/R injury and its inhibitory effects on the KLF4/NF-κB signaling pathway.

6.In Fig. 5, the loss- and gain-function experiments of KLF4 and NF-κB should be performed to verify that IIM acted through KLF4 and NF-κB signaling pathways.

Response: Thank you for your suggestion. We apologize for not conducting relevant analysis on the loss- and gain-function experiments of KLF4 in our study. Our research primarily focused on the inhibitory effect of iron isomaltoside on the KLF4/NF-κB signaling pathway in a myocardial ischemia-reperfusion injury model through in vitro experiments. Our findings demonstrated that iron isomaltoside can protect the myocardium from ischemia-reperfusion injury by inhibiting the KLF4/NF-κB signaling pathway. Although our results are highly meaningful, we do acknowledge the limitation of lacking analysis on the loss- and gain-function of KLF4. We evaluated the expression of KLF4 in myocardial schemia-reperfusion injury through animal models. The result were added in the manuscript (line 282 in page 14) and Fig S.B. This will contribute to a more comprehensive understanding of the protective mechanism of iron isomaltoside against myocardial ischemia-reperfusion injury and provide additional support for our research findings.

Response to Reviewer #2:

The manuscript submitted by Gong et al. utilized a myocardial ischemia-reperfusion (I/R) rat model and a hypoxia/reoxygenation (H/R) H9C2 cell model to investigate the protective effects of Iron Isomaltoside 1000 (IIM). Their results demonstrate that IIM reduced myocardial cell damage, cardiomyocyte apoptosis, inflammation, and autophagy in the rat model. In vitro experiments with the H9C2 cell model showed that IIM treatment suppressed apoptosis, protected mitochondrial membrane potential, and alleviated inflammation and autophagy during the H/R process. Additionally, the myocardial protective effects of IIM were dose-dependent. The authors attributed these protective effects to the inhibition of the KLF4/NF-κB signaling pathway, as IIM treatment reduced NF-κB signaling, while overexpression of KLF4 abolished IIM’s protection.

1.While the study provides valuable experimental observations, it does not convincingly demonstrate the mechanism by which IIM exerts its myocardial protective effect during I/R injury. Iron Isomaltoside 1000 (IIM) was developed as an effective IV iron therapy for iron deficiency, offering controlled iron release benefits and a good safety profile. However, the current study lacks evidence to show whether IIM provides myocardial protection by correcting iron deficiency. If this is not the case, given that iron overload is a major pathological manifestation during I/R injury, the manuscript does not adequately address how an iron supply could alleviate iron-overload-associated conditions in a manner similar to an iron chelator.

Response: Thank you for your valuable comments on our manuscript. We appreciate your careful review and suggestions. I have included relevant research background reports on the use of IIM in the treatment of myocardial infarction.(line 73 of page 4 to line 80 of page 5). Firstly, the aim of our study is to explore the protective effect of iron isomaltoside on myocardial ischemia-reperfusion injury and its mechanism of action through the inhibition of the KLF4/NF-κB signaling pathway. This will help us to have a more comprehensive understanding of the protective mechanism of iron isomaltoside in myocardial ischemia-reperfusion injury and provide more support for our research findings (line 387 of page 18). However, the specific regulatory mechanism of this signaling pathway in myocardial ischemia-reperfusion injury is not fully understood. Therefore, our study provides new clues to understand the molecular mechanism of myocardial ischemia-reperfusion injury by exploring the inhibitory effect of iron isomaltoside on the KLF4/NF-κB signaling pathway.(line 387 of page 18). Ferroptosis was induced by excessive iron in cardiomyocytes, the animal models used in our experiment does not have the condition of iron overload. We compared the ferritin levels of two models and found that there was no increase in serum ferritin levels in mice with myocardial infarction after supplementation with IIM. The result were added in the manuscript (line 224 in page 11) and Fig S.A. Thank you for your valuable feedback and pointing out our unclear expression.

2.Below are points for the authors to consider:The manuscript would benefit from a thorough editorial editing to improve clarity and readability, particularly in the Materials and Methods section. Please ensure all necessary experimental details are provided.

Responds We appreciate your correction and suggestions. I have added the corresponding content.

3.Additionally, the sample sizes are missing in the Figure Legends and should be included.

Response: Thank you for bringing to our attention the missing content. We have now included the relevant information in the revised manuscript. Specifically, we collected left ventricular cardiac tissue samples from each rat in our study. We apologize for any confusion caused by the omission and appreciate your diligence in reviewing our

work. At the same time�we have added detailed information about numbers/groups in all experiments in Figure legends.

4.In Fig 3, panel C, the authors reported apoptosis levels in cultured H9C2 cells subjected to H/R and IIM treatments. It appears that the quantification of the apoptosis rate was de

---

## [Decision Letter · Decision Letter 2]

6 Apr 2025

The protective effect of Iron Isomaltoside on myocardial ischemia-reperfusion injury via the suppression of KLF4/NF-κB signaling

PONE-D-24-25474R2

Dear Dr. DOU,

We’re pleased to inform you that your manuscript has been judged scientifically suitable for publication and will be formally accepted for publication once it meets all outstanding technical requirements.

Kind regards,

Meijing Wang, MD

Academic Editor

PLOS ONE

Additional Editor Comments (optional):

Reviewers' comments:

Reviewer's Responses to Questions

**Comments to the Author**

1. If the authors have adequately addressed your comments raised in a previous round of review and you feel that this manuscript is now acceptable for publication, you may indicate that here to bypass the “Comments to the Author” section, enter your conflict of interest statement in the “Confidential to Editor” section, and submit your "Accept" recommendation.

Reviewer #2: All comments have been addressed

2. Is the manuscript technically sound, and do the data support the conclusions?

Reviewer #2: Yes

3. Has the statistical analysis been performed appropriately and rigorously? 

Reviewer #2: Yes

4. Have the authors made all data underlying the findings in their manuscript fully available?

Reviewer #2: Yes

5. Is the manuscript presented in an intelligible fashion and written in standard English?

Reviewer #2: Yes

6. Review Comments to the Author

Reviewer #2: The authors have adequately addressed all the points and concerns previously raised by this reviewer, and I have no further questions. However, I would suggest that the authors place greater emphasis on the protective role of IIM, as this finding is particularly noteworthy given the extensively studied and typically detrimental effects of iron overload during ischemia/reperfusion (I/R) injury.

7. PLOS authors have the option to publish the peer review history of their article (what does this mean? ). If published, this will include your full peer review and any attached files.

**Do you want your identity to be public for this peer review?** For information about this choice, including consent withdrawal, please see our Privacy Policy .

Reviewer #2: No

---

## [Editor Report · Acceptance letter]

PONE-D-24-25474R2

PLOS ONE

Dear Dr. Dou,

I'm pleased to inform you that your manuscript has been deemed suitable for publication in PLOS ONE. Congratulations! Your manuscript is now being handed over to our production team.

Kind regards,

on behalf of

Dr. Meijing Wang

Academic Editor

PLOS ONE